# How Can One Strengthen a Tiered Healthcare System through Health System Reform? Lessons Learnt from Beijing, China

**DOI:** 10.3390/ijerph17218040

**Published:** 2020-10-31

**Authors:** Shuduo Zhou, Jin Xu, Xiaochen Ma, Beibei Yuan, Xiaoyun Liu, Hai Fang, Qingyue Meng

**Affiliations:** 1School of Public Health, Peking University, Beijing 100191, China; zhoushuduo@pku.edu.cn; 2China Center for Health Development Studies, Peking University, Beijing 100191, China; xujin@bjmu.edu.cn (J.X.); xma@hsc.pku.edu.cn (X.M.); beibeiyuan@bjmu.edu.cn (B.Y.); xiaoyunliu@pku.edu.cn (X.L.); hfang@hsc.pku.edu.cn (H.F.)

**Keywords:** health system reform, tiered healthcare system, China

## Abstract

How one can reshape the current healthcare sector into a tiered healthcare system with clarified division of functions between primary care facilities and hospitals, and improve the utilization of primary care, is a worldwide problem, especially for the low and middle-income countries (LMICs). This paper aimed to evaluate the impact of the Beijing Reform on healthcare-seeking behavior and tried to explain the mechanism of the change of patient flow. In this before and after study, we evaluated the changes of outpatient visits and inpatient visits among different levels of health facilities. Using the monitored and statistical data of 373 healthcare institutions 1-year before and 1-year after the Beijing Reform, interrupted time series analysis was applied to evaluate the impact of the reform on healthcare-seeking behavior. Semi-structured interviews were used to further explore the mechanisms of the changes. One year after the reform, the flow of outpatients changed from tertiary hospitals to community health centers with an 11.90% decrease of outpatients in tertiary hospitals compared to a 15.01% increase in primary healthcare facilities. The number of ambulatory care visits in primary healthcare (PHC) showed a significant upward trend (*P* < 0.10), and the reform had a significant impact on the average number of ambulatory care visits per institution in Beijing’s tertiary hospitals (*p* < 0.10). We concluded that the Beijing Reform has attracted a substantial number of ambulatory care visits from hospitals to primary healthcare facilities in the short-term. Comprehensive reform policies were necessary to align incentives among relative stakeholders, which was a critical lesson for other provinces in China and other LMICs.

## 1. Introduction

It is a global consensus that chronic disease is the main driver of mortality and morbidity and has the highest costs in the context of aging, multi-morbidity and complexity. A high-cost specialist or hospital-centered health system cannot effectively address the non-communicable disease (NCD) epidemic. Primary healthcare (PHC) is the foundation of a sustainable healthcare system [1]. However, it is a worldwide problem that patients increasingly choose to access already severely overcrowded high-level hospitals, leaving lower level facilities with low utilization rates. The utilization of primary healthcare in many countries in Sub-Saharan Africa has been limited, and those who bypassed lower levels of healthcare cited the quality of care, the standard of healthcare providers, and previous positive experiences as significant reasons for seeking care directly at the hospital [2,3]. In India, the utilization of basic health services has remained poor and the utilization of primary health centers is found to be less [4]. Additionally, to improve the use of primary health services, the current health sector reform in South Africa focuses on strengthening the formalization and integration of community-based services in the district and sub-district level [5]. Similarly, high-income countries tried to strengthen primary care for cost containment, meeting the needs of the aging population and improvement of health equity [6]. The share of general practitioners among all physicians dropped from 32% in 2000 to 29% in 2016 across all OECD countries [7]. Underutilization of primary care and the shortage of young primary care practitioners in primary care are also thorny problems facing the United States [6,8].

Strengthening the primary care system, promoting utilization of primary care, and establishing a reasonable hierarchical medical care system are the major targets of China’s new health system reform [9,10]. The Chinese central government has encouraged local government to pilot innovative interventions to gain experience for the complete health system reform. In the first phase of the new health system reform, the main tasks were establishing a sound mechanism for basic medical and healthcare and strengthening said endeavor at the community level. All levels of government have poured resources into primary care infrastructure construction, personnel training, and other supporting programs, including salary reforms, a zero-profit drug policy, and insurance schemes at the primary care level [11]. However, the investment does not seem to lead to a successful tiered healthcare system with more people seeking medical services in hospitals rather than primary care institutions [12,13,14]. According to the National Health Commission Statistical Yearbook 2009–2019, during the period 2009 to 2019, the proportion of PHC workers, the proportion of PHC outpatient visits, and the proportion of PHC hospitalizations decreased by 8.28%, 8.78%, and 13.82% respectively. In contrast, the proportion of tertiary health workers, proportion of visits at the tertiary hospitals, and proportion of hospitalization at the tertiary hospitals increased by 9.90%, 9.77%, and 16.38%, respectively. The reasons for the phenomena are quite complex, and health knowledge, patient habits, drug variety, medical equipment, and service portfolios are the main factors influencing choices of healthcare facilities in China [15]. In addition, some researches pointed out that disease severity, medical staff, equipment, and drug availability played important roles for people when choosing healthcare facilities, and improvements in drug availability, medical staff’s skills, and equipment in PHC could redirect the patient flow from higher level hospitals to PHC [16,17]. 

How to redirect the patient flow from tertiary hospitals to PHC providers is a key for a more effective healthcare delivery system. However, how to build a reasonable tiered healthcare system with clarified division of functions between primary care facilities and hospitals still lacks specific evidence to guide the way [18]. In April 2017, Beijing’s municipal government initiated a comprehensive public hospital reform of separating drug sales from hospital revenues (the Beijing Reform) to reduce the rapid increase of medical expenditure and to build a reasonable hierarchical medical care system with all the public health facilities involved in this reform (Table 1). The main reform strategies are as the follows: firstly, a hierarchical medical service fee replaced the 15% drug markups, registration and treatment fees in all public healthcare facilities; secondly, to encourage the use of primary care services, community medical institutions were given the same access to the medicines that were usually prescribed in high-level hospitals and the long prescription policy (once for two months) was designed for chronic patients who were eligible; thirdly, price adjustment—prices of 435 medical service items were adjusted, with increased prices for surgical operations and traditional Chinese medicine services and decreased prices for diagnostic tests (for example, computed tomography and magnetic resonance imaging) [19]. The hierarchical medical service fee increased the “admission price” to medical treatment in higher level hospitals, especially for patients not covered by Beijing health insurance scheme [20]; meanwhile, the change of the drug catalogue improved the accessibility of drugs in primary health institutions. Thus, the implementation of the Beijing Reform, especially the cancellation of the drug markups, the establishment of hierarchical medical service fees, and the improving availability of medicines for chronic patients, should have certain impacts on patients’ healthcare-seeking behaviors.

As a key part of the new health reforms in China that started in 2009, since its launch in April 2017, the Beijing Reform attracted nationwide attention from the central and local government, and civil society. However, there have been few empirical studies examining the overall impact of the Beijing Reform on the primary healthcare utilization and patients’ healthcare-seeking behaviors, especially for further explaining the mechanisms of change for patient behaviors. This study aims to evaluate the impacts of the Beijing Reform on healthcare-seeking behavior and tries to explain the mechanism of the change for patient flow. This study will contribute to the policy development and literature database in the following aspect. Through policy evaluation, we will further improve the follow-up policy design, provide policy recommendations to ensure stable and efficient operation of the Beijing health system, and summarize lessons for other low and middle-income countries (LMICs).

## 2. Methods

### 2.1. Data Sources 

The main data used in this study are from the monthly monitored data from 373 public medical institutions (including all of the secondary and tertiary public hospitals and over 60% of the total community health centers in Beijing). A hospital in Beijing can be owned by the city, ministries, or large enterprises, and all of them are included in the monitored lists. Table 2 shows the specific classification and quantity of the monitored facilities (Table 2). Our data covered a 1-year period before and 1-year period after the Beijing Reform in April 2017. All the monitored facilities were required to regularly report on a set of key indicators, including expenditures and service volumes, to the Municipal Health Commission. All data were collected from Beijing Municipal Health Commission Information Center with permission. On the other hand, to further explain the mechanism of change for patient healthcare-seeking behavior, two rounds of semi-structured key informant interviews were conducted three months and one year after the reform, respectively. The keywords of the interviews focused on the changes of the patients’ healthcare-seeking behaviors, and the design and implementation of the Beijing Reform. Considering the principles of representativeness and information saturation, qualitative interviewees mainly involved four types of stakeholders, including government agencies, medical institution administrators, medical staff, and patients from all levels of healthcare facilities which covered the design, implementation process, and results of the Beijing Reform. For each type of interviewee we interviewed fifteen people, and the total number of interviewers was about sixty for each round of interview.

### 2.2. Measurement

To analyze the impact of the Beijing Reform on the patients’ selection of different levels of medical institutions, the flow of outpatients was measured. To ensure comparability, we summed the service volumes from each level of institution and divided that by the number of institutions. The indicators included the absolute monthly outpatient visits per primary health center, per secondary hospital, and per tertiary hospital, and the absolute monthly inpatient hospitalizations per secondary hospital and per tertiary hospital. We expect that the Beijing Reform should increase the outpatient and inpatient visits in primary health centers and reduce the outpatient and inpatient visits in secondary or tertiary hospitals; meanwhile, the proportion of outpatient volume in primary health centers should be higher compared with before. Meanwhile, we assumed that the number of inpatient visits would have had no significant change.

### 2.3. Data Analysis

Descriptive statistical analysis was used to pre and post-reform data. In addition, interrupted time series analysis was used to assess the trends and levels changes in the number of outpatient visits and number of inpatient hospitalizations in all levels of healthcare facilities. Specifically, we analyzed the trends and levels changes of various indicators before and after the reform from April 2016 to March 2018. The interrupted time series model used in this study is specified as the following function:(1)$$\gamma_{t}=\beta_{0}+\beta_{1}\times \mathrm{time}+\beta_{2}\times \mathrm{intervention}+\beta_{3}\times \mathrm{time}\;\mathrm{after}\;\mathrm{intervention}+\beta_{4}\mathrm{Xi}+\epsilon$$
where *γ_t_* refers to the indicators at month t; *time* is a continuous time variable from the beginning of the observation period to the month of time t; *intervention* means that intervention is implemented from time t, 0 before the intervention and 1 after the intervention; the *time after intervention* is an interaction term. *β_0_* is the estimated value of the baseline level of the result; *β_1_* is the estimated value of the baseline trend; *β_2_* is the estimated value of the instantaneous change of the indicator after the intervention; *β_3_* is the estimated value of the slope difference before and after the intervention; *β_4_* represents the seasonality effect, and we use the Fourier terms (pairs of sine and cosine functions) to control for the seasonality and other long term trends [21]; *ε* represents the error term. The autocorrelation problem is common in time series data and may cause the model estimation parameters to be invalid. In order to solve this problem, this study adopted the robust small sample Cumby–Huizinga autoregressive test, and selected the statistically significant minimum order by the statistical test of the regression lag order as the order of the Newey-West regression standard error to eliminate the hazards of autocorrelation to the model [22,23]. A *p* value < 0.10 was considered statistically significant. Stata V.14.1 for Windows (Stata Corp, College Station, Texas, USA) was used for the statistical analysis.

For qualitative analysis, the consolidated criteria for reporting qualitative studies (CQREQ) checklist was completed through the research team, study design, and analysis and findings [24]. The research team were from the Chinese Center for Health Development and Studies. Each of the interviewers was trained before the interview. The average time of the semi-structured interviews was about forty minutes for the interviewees. After obtaining the consent of the interviewee, the video recording was made and the text was transcribed verbatim. Theoretical coding was performed using NVivo 11.0 (QSR International, Melbourne, Australia) [25]. Two well-trained authors (S.D.Z., J.X.) were recruited to transcribe and code the audio recordings. Codes were improved through pilot tests and discussion, resulting in a set of four final descriptive categories.

### 2.4. Patient and Public Involvement

Patients were not involved in the design or management of this research. We used data from the Beijing Municipal Health Commission Information Center, and the data were collected at the institutional level with no direct involvement of patients.

## 3. Results

### 3.1. Impacts of the Beijing Reform on the Healthcare-Seeking Behavior of Outpatients

One year after the implementation of the reform, the average number of outpatients per institution in the monitored PHC institutions showed a clear upward trend compared with the same period of last year. The average number of outpatients per PHC institution was 117.6 thousand person-times, an increase of 15.01% compared with the year before the Beijing Reform. In the tertiary hospitals, the average number of outpatients per institution was 1094.7 thousand person-times, a decrease of 11.90% compared with the same period of last year, while the outpatients in the secondary hospitals had no significant change. In the year before the reform, the total number of outpatients of all monitoring units was 162.2 million, of which the total number of outpatients in PHC institutions was 21.1 million, and the proportion was 12.99%. One year after the reform, the total number of outpatients of all monitoring units was 152.2 million, of which the number of outpatients in primary healthcare institutions was 24.2 million, and that the proportion was 15.91% (Table 3).

The PHC institutions are mainly designed for the outpatient services in Beijing, and almost all the PHC institutions have no hospitalization beds. We analyzed the changes of inpatient services in secondary and tertiary hospitals. One year after the reform, the number of inpatient visits per secondary hospital was 6939.07 person-times, with a 0.64% decrease compared with the previous year. The number of inpatient visits for per tertiary hospital was 31851.97 person-times, with a 2.67% increase compared with the previous year (Table 4).

### 3.2. Changes of Outpatient Flow between PHC Providers, Secondary Hospitals, and Tertiary Hospitals

Interrupted time series analysis (ITS) was used to estimate the impacts of Beijing Reform on the level and trend changes in the monthly number of ambulatory care visits per primary healthcare institution by month. The results showed that after the reform, the number of ambulatory care visits decreased by 938.82 person-times (*p* = 0.58). The monthly ambulatory care visits before the reform was 135.19 person-times/months (*p* = 0.38); the reform had increased the monthly number outpatient visits by 108.46 person-times/month (*p* = 0.22). After the reform, the number of ambulatory care visits in PHC showed a significant upward trend (*p* < 0.10) (Figure 1, Table 5).

Figure 2 displays the TS analysis of the impacts of Beijing Reform on the level and trend changes in the monthly number of ambulatory care visits per secondary hospital by month. The results showed that after the reform, the number of ambulatory care visits decreased by 5453.08 person-times (*p* = 0.36); the monthly increase of ambulatory care visits before the reform was 367.15 person-times/month (*p* = 0.50), and the monthly increase of ambulatory care visits after the reform increased to 590.30 person-times/month (*p* = 0.22); the reform increased the monthly ambulatory care visits by 223.15 person-times/month (*p* = 0.50). The results show that the Beijing Reform had no significant influence on the outpatient visits in secondary hospitals (Figure 2, Table 5).

Figure 3 shows the ITS analysis of the impacts of Beijing Reform on the level and trend of changes in the monthly number of ambulatory care visits per tertiary hospital by month. The results showed that after the reform, the number of ambulatory care visits decreased by 27,423.35 person-times (*p* < 0.10); the monthly increase of ambulatory care visits before the reform was 1293.70 person-times/month (*p* = 0.33), and the monthly increase of ambulatory care visits after the reform decreased to 1215.50 person-times/month (*p* = 0.33); the reform had reduced the monthly increase of ambulatory care visits by 78.22 person-times/month (*p* = 0.92). This shows that the reform had a significant impact on the average number of ambulatory care visits per institution in Beijing’s tertiary hospitals (*p* < 0.10), leading to the average number of ambulatory care visits in tertiary hospitals reducing significantly (Figure 3, Table 5).

### 3.3. Changes of Inpatient Flow between PHC Providers, Secondary Hospitals, and Tertiary Hospitals

Figure 4 and Figure 5 describe the ITS analysis of the impacts of Beijing Reform on the level and trend changes in the number of inpatient visits per tertiary and secondary hospital, respectively. The results demonstrate that after the reform, the number of inpatient visits decreased by 52.15 person-times (*p* = 0.50) for per secondary hospital, and the number of inpatient visits decreased by 333.86 person-times (*p* = 0.43) for per tertiary hospital. The reform reduced the monthly increase of ambulatory care visits by 2.83 person-times/month (*p* = 0.44) per secondary hospital and the reform reduced the monthly increase of ambulatory care visits by 1.36 person-times/month (*p* = 0.94) per tertiary hospital, (Figure 4 and Figure 5, Table 6)

### 3.4. Explanations for Changes of the PHC Utilization from Interviewees

Three main themes emerged from qualitative interviews.

### 3.5. The Main Reasons for Increase in Ambulatory Care Visits for PHC Institutions

The increase in ambulatory care visits at primary healthcare institutions is one of the highlights of this reform, and the reason for the increase is also worthy of further analysis. In qualitative interviews, we found that the vast majority of respondents believed that the increase in the number of ambulatory care services at primary medical institutions was due to the guidance of medical service fees (including the policy of exempting medical service fees for seniors over 60 years of age) and the docking of the drug catalogues and long prescriptions for four chronic diseases (hypertension, diabetes, stroke, and coronary heart disease).

As for a secondary or tertiary hospital, now you have to pay at least 20 yuan, 40 yuan, and 60 yuan for the deputy chief physicians or experts. Then if he is a minor patient, he won’t go. If he is over 60 years old, he doesn’t need to spend money on medical service fee in PHC. The change is very obvious and many people have turned around to the community health centers. (Director of a community health service center, July 2017).

Additionally, we have docking 105 catalogues and prescriptions for four chronic diseases (hypertension, diabetic stroke, coronary heart disease) between community health centers and big hospitals which is very important. The long prescription policy is also very helpful. First you need to sign a contract, if the diagnosis is clear and the condition is relatively stable, and it is suitable for long prescriptions at home. (Community Health Management Center of Beijing Municipal Health Commission, June 2016).

In addition, “synchronization of reform and service improvement” is a major feature of this reform. Medical institutions, especially the primary healthcare institutions, are actively responding to the specifications, and have adopted some measures to improve the “feelings” of patients, mainly focusing on the improvement of the consultation process and the environment.

We have done a few things to improve. For example, to save the time and improve convenience, the community healthcare centers explore a convenient way for patients to pay, that is, after you see doctor, the payment will be completed once. You pay the drug inspection fee and medical service fee once. (Community Health Management Center Beijing Municipal Health Commission, June 2016).

In fact, it means that we still want to increase the sense of experience or gain of the patients. We increased the green plants, then purchased the waiting chairs in our lobby, and the pharmacy added this automatic medicine dispenser. (Director of a community health service center, July 2017).

### 3.6. The Main Reasons for the Decrease in Ambulatory Care Visits for Tertiary Hospitals

Regarding the decrease in the number of ambulatory care patients in tertiary hospitals, the interviewees had relatively consistent conclusions in the qualitative interviews. Everyone agreed that the increase in medical service fees was the main reason for the decline in the number of emergency visits. The number of patients (mainly for prescribing drugs) and the number of minor patients, such as those coming in for hypertension, decreased.

I think for the decline in the number of outpatients, and the most important reason is the impact of the medical service fee. (Doctor of Oncology, a third-level hospital, July 2017)

Outpatients are definitely much less than before, especially those in our department, because our department has some chronic diseases, and they come regularly to get medicines. Actually for this ten yuan for medical service, the old man cares very much, so obviously less than before. (Doctor of Cardiology, a tertiary hospital, June 2018)

### 3.7. The Implementation Process of the Beijing Reform

Meanwhile, this reform implemented a detailed and systematic promotion plan, with particular emphasis on ensuring that all relevant institutions and personnel were aware of their respective responsibilities through training and publicity, and ensuring that the policies are accurately transmitted to all people.

Before the policy appeared, the government departments learned and mastered in advance and then training the medical institutions, training the chief dean, the deputy dean, the main leadership team. The next step was the gradual training at the hospital level, and everyone has been trained. (The Beijing Municipal Health Commission, July 2017)

The first was to organize special training in the hospital to give training to each department, and then the department and the head nurse gave us training for doctors. The reform policy was not only for patients to understand, we also needed to understand before we could explain to the patients. We had been trained for a long time. (Physician at a tertiary Hospital, July 2017)

## 4. Discussions

The study found that the Beijing Reform had a significant impact on the flow of outpatients, while the flow of inpatients remained as before. The number of outpatients in tertiary hospitals decreased significantly; meanwhile, the number of outpatients in community health centers increased. It is worth noting that the number of outpatients to the community health centers showed a clear upward trend after the reform. The inpatient visits for tertiary or secondary hospitals had no significant changes, which were in accordance with our hypothesis, as the Beijing Reform had no direct policies for the inpatient services. These pieces of evidence indicate that the Beijing Reform has pushed outpatients to seek healthcare from PHC providers, which is critical for a tiered healthcare delivery system. The case from Beijing provides a good example for how a combined reform of pricing policy and improved availability of PHC can work for strengthening the health system.

From the comparative analysis: the reason that the Beijing Reform played an important role in changing the flow of outpatients is that the designed comprehensive policies increased the attraction of PHC institutions for the mild patients. In China, the PHC staff often lack full medical training and are licensed as assistant doctors and serve as general practitioners, while the staff in secondary or tertiary hospitals are generally fully qualified and serve as specialists [26]. The hierarchical medical service fee, the improving availability of drugs, and the convenience and time savings brought by long prescription are major factors leading to the change of patient flow. The medical service fee for an outpatient visit was raised from Y5 to Y50 in tertiary hospitals, Y30 in secondary hospitals, and Y20 in primary healthcare facilities, and the patient’s co-payment on medical service fees are Y10, Y2, and Y1, respectively. The demand for specialists had relative high elasticity [27], so the increased price decreased the utilization of higher-level services for the minor patients, which was confirmed by the results of key informant interviews. The price signal guided the patients to scientifically and rationally choose the type of doctor to visit [28]. On the other hand, the expansion of essential drug catalogues may have improved the accessibility of primary drug products, and drug accessibility is one of the important factors affecting the service utilization of primary medical institutions [17,29]. Meanwhile, for the chronic disease patients, whose diagnoses are clear and whose conditions are relatively stable, long prescriptions are suitable (once for two months). One year after the reform, long prescriptions were signed by more than 10,000 people—more than 40,000 prescriptions. The policy was also encouraged as a pilot from the central government to direct patient flow toward primary care [30], and the effectiveness was confirmed by previous studies [16]. The geographical availability and the availability of medicines have enabled patients to return to the grassroots, especially for chronically ill patients.

In addition, multiple supporting policies were introduced to match the comprehensive healthcare reform, and this reform introduced the slogan of “synchronization of reform and service improvement.” Additional policies were implemented as the supporting plan. Firstly, elderly people over the age of 60 can enjoy the policy of exemption from medical service fees when they seek medical treatment at the primary healthcare institutions. Secondly, the service model of the pre-medical and post-paying services in more than 200 community health service centers has shortened the time and improved convenience for patients. Thirdly, the environment of primary healthcare centers was improved to make patients feel more comfortable when seeing doctors. All the measures had led to an increase in the number of outpatients in primary healthcare institutions.

The well-organized leadership, monitoring, and promotion mechanisms played vital roles in the implementation of the reform, which guaranteed the reform achieved the goals as it planned. The power of policy enforcement determines the public awareness level and policy effectiveness, as shown in previous studies [31,32]. As for the Beijing Reform, the Deputy Prime Minister of the State Council personally served as the “General Leader” of the Beijing Reform, and established the Beijing Reform Coordination Group across the central ministries and provincial governments. The Secretary of the Beijing Municipal Party Committee co-chaired with the director of the National Health Commission as a “double team leader.” The competent authorities in the health administration devoted greater efforts to policy promotion to inform the public and the relevant stakeholders about the comprehensive reform policy. This reform implemented a detailed and systematic promotion plan, with particular emphasis on training and publicity to ensure that all relevant institutions and personnel were aware of their respective duties. Layer-by-layer training makes policies well known to all stakeholders.

The Beijing Reform set a good example for the whole country and other low and middle-income countries in how to direct the patients to the PHC and improve the efficiency of the whole health system. The experience of Beijing Reform showed that the prices of medical service fees, the availability of drugs, and the improvement of convenience had significant impacts on patients’ healthcare-seeking behaviors, and the factors could be important knobs for future policy design. Based on this, in the future policy design of public hospital reform, what we should remember is that the healthcare system is an interdependent macrocosm, and systematic thinking is of great importance to maximize the policy’s effect through an organic policy combination.

The increase in the number of outpatients has led to a significant increase in technical efficiency of the primary healthcare institutions, but more attention should be paid to the following aspect to establish an effective tiered service delivery system: the relatively low quality of the primary healthcare, which has not been solved through the Beijing Reform. Through this reform, the price and convenience were two major factors to direct the patients to community health centers; however, the quality of primary healthcare is one of the most important factors affecting patient healthcare-seeking behavior [33,34]. Further policies are needed in the following aspects. Firstly, strengthening the capacity of community health service institutions through improving the quality and quantity of human resources. Secondly, establishing a family doctor contracting system, to better cope with the increase in the outpatients of primary institutions due to the reform, and to promote the advancement of tiered healthcare system. Thirdly, making innovations and breakthroughs in the reform of the basic-level salary system, effectively increasing the income of primary-level health workers, and improving the ability of primary medical institutions to attract and retain the necessary healthcare providers.

There are some limitations of the current study. First, due to the short reform period since the policy implementation, this study only provides proximal effects of the reform, which might affect the robustness of our analysis and conclusion. Second, the results presented in the study are the combined effects of the comprehensive policies, so the effects of every individual policy are not clear. Third, because of the lack of individual outpatient data, we could not check the changes in the specific departments visited by patients with NCDs. Fourth, there are more healthcare providers in the PHC of Beijing compared with the western or other less developed regions in China, so we should be cautious when using the lessons learnt herein in those regions. Further studies could focus on the contributions of a single policy and explore the outpatient changes with NCDs among different levels of healthcare facilities in the future.

## 5. Conclusions

The reform has directed the flow of patients to primary health facilities. With the flow of outpatients being more reasonable, the allocative efficiency of the health system has been improved to some extent. The Beijing Reform demonstrates a systemic reform can build a reasonable tiered healthcare system. Comprehensive policy and an effective leadership, monitoring, and promotion mechanism is necessary for the success of changes. Further reform is needed to strength the capacity of primary medical institutions and improve the incentives for the staff in primary medical institutions

## Figures and Tables

**Figure 1 ijerph-17-08040-f001:**
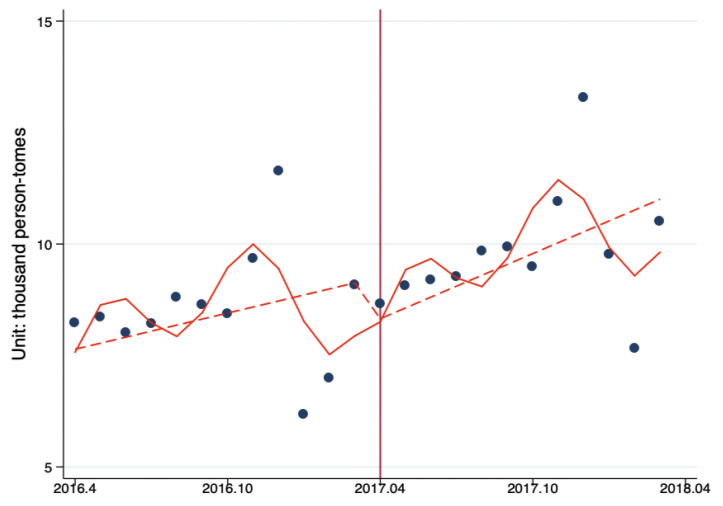
The ITS analysis of monthly outpatient visits in per primary healthcare (PHC) institution. Model adjusted for seasonality and autocorrelation. Solid line: Predicted trend based on the seasonally adjusted regression model. Dashed line: De-seasonalized trend. Vertical line: Intervention began. Blue dots: Actual number.

**Figure 2 ijerph-17-08040-f002:**
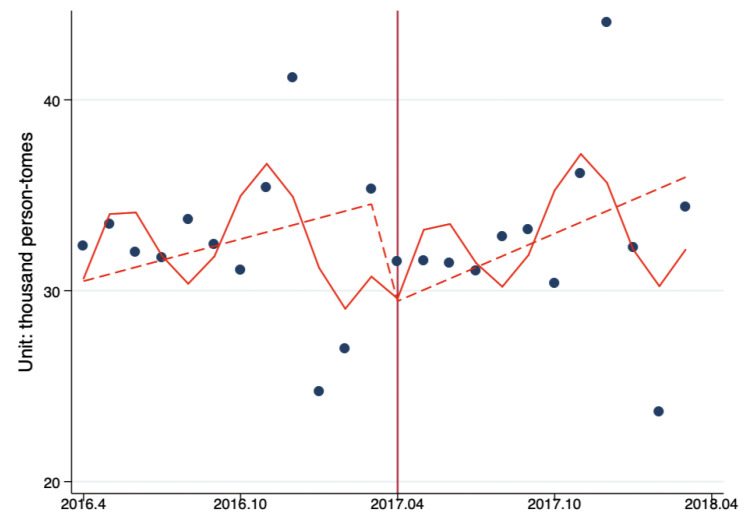
The ITS analysis of monthly outpatient visits per secondary hospital. Model adjusted for seasonality and autocorrelation. Solid line: predicted trend based on the seasonally adjusted regression model. Dashed line: de-seasonalized trend. Vertical line: intervention began. Blue dots: actual number.

**Figure 3 ijerph-17-08040-f003:**
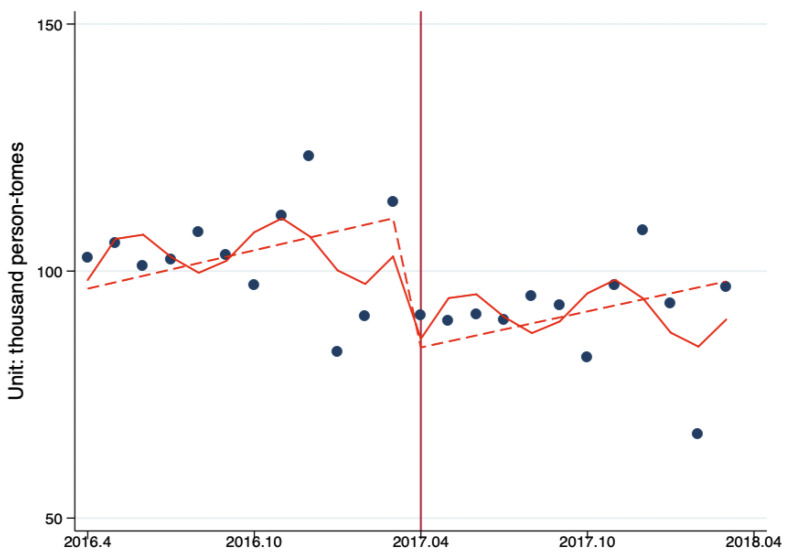
The ITS analysis of monthly outpatient visits per tertiary hospital. Model adjusted for seasonality and autocorrelation. Solid line: predicted trend based on the seasonally adjusted regression model. Dashed line: de-seasonalized trend. Vertical line: intervention began. Blue dots: actual number.

**Figure 4 ijerph-17-08040-f004:**
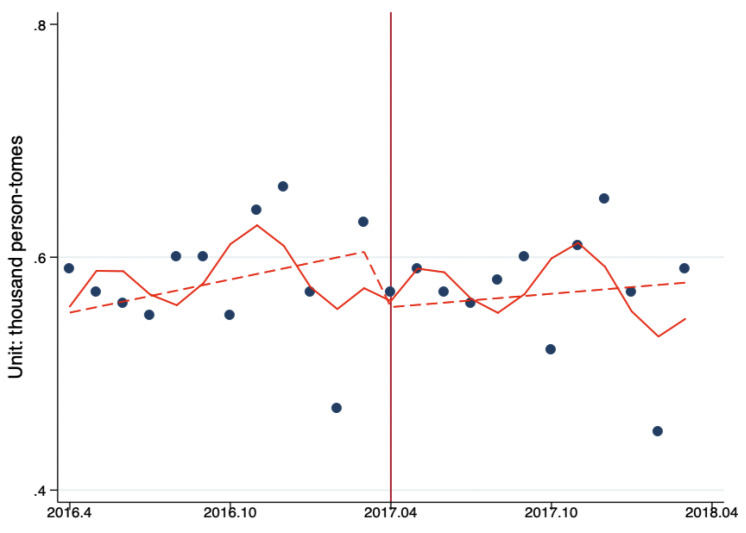
The ITS analysis of monthly inpatient visits in per tertiary hospital. Model adjusted for seasonality and autocorrelation. Solid line: predicted trend based on the seasonally adjusted regression model. Dashed line: de-seasonalized trend. Vertical line: intervention began. Blue dots: actual number.

**Figure 5 ijerph-17-08040-f005:**
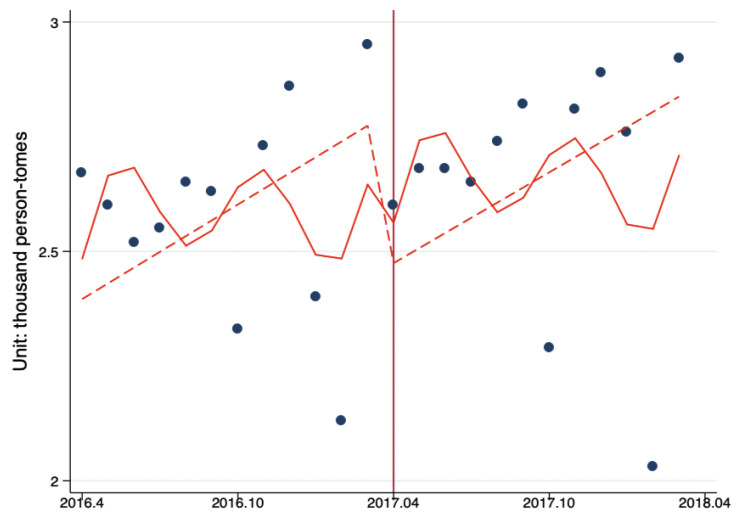
The ITS analysis of monthly inpatient visits per secondary hospital. Model adjusted for seasonality. Solid line: predicted trend based on the seasonally adjusted regression model and autocorrelation. Dashed line: de-seasonalized trend. Vertical line: intervention began. Blue dots: actual number.

**Table 1 ijerph-17-08040-t001:** The details of the main strategies for the Beijing Reform in April 2017.

Reform Measures	Descriptions of Reform Measures
Zero mark-up of drug sales and hierarchical medical service fee	15% mark-up of drug sales were removed in all public healthcare facilities; A hierarchical medical service fees with higher level hospitals and senior physicians charging higher service fees; The junior service fees was increased from Y3.5 toY20 in PHC, from Y4 to Y30 in secondary and from Y5 to Y50 in tertiary, and for senior physicians, the new medical service fee is Y60, Y80, Y100 and Y50, Y70,Y90 for tertiary and secondary, respectively; Out-of-pocket expenditures were reimbursed by the Beijing Health Insurance with Y19, Y28, Y40 for primary, secondary and tertiary, respectively.
Changes of drug catalogues in PHC	105 kinds of drugs for chronic diseases are available at PHC, and the drug catalogues are the same as the secondary or tertiary hospitals The long prescription policy is designed for chronic patients with two months prescription for once.
Prices adjustment of 435 medical service items	Prices for surgical operations and traditional Chinese medicine services are increased. Prices for diagnostic tests (CT and MRI) are decreased. All the services changes are covered by the Beijing Health Insurance, and the Out-of-pocket expenditures remain the same.

**Table 2 ijerph-17-08040-t002:** The levels and classifications of monitored public medical institutions.

Levels	Classifications	Numbers
Tertiary hospitals		89
	Traditional Chinese Medicine Hospitals	23
	Specialist Hospitals	20
	General Hospitals	46
Secondary hospitals		78
	Maternal and Child Health Hospitals	13
	Traditional Chinese Medicine Hospitals	12
	Specialist Hospitals	11
	General Hospitals	42
Primary health institutions		206
In total		373

**Table 3 ijerph-17-08040-t003:** The outpatient changes at medical institutions of each level (thousand person-times).

Medical Institutions	Year	Second Quarter	Third Quarter	Fourth Quarter	First Quarter *	Total
Tertiary hospitals	2016	309.2	313.4	331.5	288.5	1242.6
	2017	272.1	278.0	287.7	256.9	1094.7
	Percent increase	−12.01%	−11.29%	−13.23%	−10.94%	11.90%
Secondary hospitals	2016	97.8	97.9	107.6	87.0	390.2
	2017	94.5	97.1	110.5	90.3	392.4
	Percent increase	−3.35%	−0.79%	2.72%	3.79%	0.55%
Primary health centers	2016	24.6	25.7	29.7	22.2	102.2
	2017	26.9	29.0	33.7	27.9	117.6
	Percent increase	9.40%	13.11%	13.35%	25.60%	15.01%

***** The figures of first quarter for the 2016 line are actually for the first quarter of 2017; the 2017 line refers to the first quarter of 2018.

**Table 4 ijerph-17-08040-t004:** The inpatient changes at each level of medical institution (person-times).

Medical Institutions	Year	Second Quarter	Third Quarter	Fourth Quarter	First Quarter *	Total
Tertiary hospitals	2016	7787.46	7827.49	7922.44	7486.42	31,023.81
	2017	7951.04	8207.90	7986.67	7706.35	31,851.97
	Percent increase	2.10%	4.86%	0.81%	2.94%	2.67%
Secondary hospitals	2016	1718.15	1748.47	1846.82	1670.46	6983.90
	2017	1718.69	1738.51	1771.39	1710.48	6939.07
	Percent increase	0.03%	−0.57%	−4.08%	2.40%	−0.64%

***** The figures of first quarter for the 2016 line are actually for the first quarter of 2017; the 2017 line refers to the first quarter of 2018.

**Table 5 ijerph-17-08040-t005:** The impacts of the Beijing Reform on monthly outpatient visits per healthcare facility type.

Level/Trend	PHC	Secondary	Tertiary
Baseline trend	0.135 (0.149)	0.367 (0.534)	1.294 (1.300)
Level change	−0.939 (1.656)	−5.453 (5.729)	−27.423 (14.358) *
Trend change	0.108 (0.084)	0.223 (0.322)	−0.078 (0.775)
Post-reform trend	0.244 (0.139) *	0.590 (0.458)	1.216 (1.201)

The model was adjusted for seasonality and autocorrelation with the Fourier terms and Newey-West for regression and standard error, respectively. * *p* < 0.10.

**Table 6 ijerph-17-08040-t006:** The impacts of the Beijing Reform on monthly inpatient visits per healthcare facilities.

Level/Trend	Secondary	Tertiary
Baseline trend	0.005 (0.007)	0.034 (0.037)
Level change	−0.052 (0.076)	−0.334 (0.410)
Trend change	−0.003 (0.004)	−0.001 (0.019)
Post-reform trend	0.002 (0.006)	0.033 (0.032)

The model was adjusted for seasonality and autocorrelation with the Fourier terms and Newey-West regression and standard error, respectively. * *p* < 0.10.

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
