# Peer review of "How Can One Strengthen a Tiered Healthcare System through Health System Reform? Lessons Learnt from Beijing, China"

_ijerph, 2020, doi:10.3390/ijerph17218040_

Round 1
Reviewer 1 Report
Dear authors, I would like to congratulate your work and I hope your manuscript could be read shortly.
I would like to recommend some points to change in your work:
Line 77: Please check the space before parentheses.
Line 85: Please consider the substitution of medical investigations for diagnostic tests and check the double parenthesis inline 86.
Line 93: Table 1, on the same page, and consider aligning the bullet point to the left. Divide the table in three and mark the year of implementation.
Line 110: please check the space before parentheses.
Line 158: please cite Nvivo Program: NVivo qualitative data analysis software; QSR International Pty Ltd. Version 11, released 2015.
This is a quantitative-qualitative study and the authors present some verbatim from interviews. I think this is a good point and deserves more consideration. From interviews authors establish conclusions and maybe these conclusions need further interpretation. Discourse saturation points, keywords, and consideration for all intervenient professionals could be pointed out.
About the reference list I have some doubts:
Line 432: Reference number 4: this is a book? a report? editorial?
Thank you very much. I appreciate your work.
Author Response
Response to reviewers
Dear editor,
Thank you very much for organizing the comments on our manuscript. We have revised the manuscript according to the editorial requirements and the comments from reviewers. Please find the responses to the Comments and Editorial Requirements below.
Comments:
-Dear authors, I would like to congratulate your work and I hope your manuscript could be read shortly.
Authors’ response: Thank you for your kind reminders. We have done proofreading and drop the reduplicate content of the interviews.
-Line 77: Please check the space before parentheses.
-Line 85: Please consider the substitution of medical investigations for diagnostic tests and check the double parenthesis inline 86.
-Line 93: Table 1, on the same page, and consider aligning the bullet point to the left. Divide the table in three and mark the year of implementation.
-Line 110: please check the space before parentheses.
Authors’ response: Thank you for your kind reminders. We have done proofreading and revised as your suggestions.
-Line 158: please cite Nvivo Program: NVivo qualitative data analysis software; QSR International Pty Ltd. Version 11, released 2015.
Authors’ response: Thank you for your helpful suggestion. We have added the reference in the article.
-This is a quantitative-qualitative study and the authors present some verbatim from interviews. I think this is a good point and deserves more consideration. From interviews authors establish conclusions and maybe these conclusions need further interpretation. Discourse saturation points, keywords, and consideration for all intervenient professionals could be pointed out.
Authors’ response: Thank you for your comments and helpful suggestions. The conclusions from the interviews were mainly interpreted in the discussion section, for example the second, third and fourth paragraphs of the discussion. Discourse saturation points, keywords, and consideration for all intervenient professionals have been added in the Methods section(line: 120-126)
-About the reference list I have some doubts: Line 432: Reference number 4: this is a book? a report? editorial?
Authors’ response: Line 432: Reference number 4 is an article of the
International Journal Of Community Medicine And Public Health

Reviewer 2 Report
This paper looks as a social experiment in relationship between health policy and economy.
This paper shows the health system is a interdependency macrocosm and an organic policy combination can change totally the economy of this system. It is a starting point for a future long term analysis.
Because the great number of patients in local Health Care System, really this paper looks very interesting.
The article is well written and explained. When I speak in my comment about “any ethic consideration” I mean that in the paper the authors don’t take in consideration the real satisfaction of the patients who are the protagonists of the study. When a group of scientists decide to front a political/social experiment, I suppose, they have to value also the impact they can reach on human emotionality and not only on the economy of the system. But in the last times, we are living, this is considered really not so important respect the reached good result in economy and efficiency. The question is: what we are really targetting in health policy for first? Good results in economy or good results in human satisfaction?
So this paper, I repeat good explained and written, make to take in ethic consideration this last aspect despite the well done job from sociological/scientific point of view.
I think it can be published as it is.
Author Response
Response to reviewers
Dear editor,
Thank you very much for organizing the comments on our manuscript. We have revised the manuscript according to the editorial requirements and the comments from reviewers. Please find the responses to the Comments and Editorial Requirements below.
Comments:
This paper looks as a social experiment in relationship between health policy and economy.
This paper shows the health system is a interdependency macrocosm and an organic policy combination can change totally the economy of this system. It is a starting point for a future long term analysis.
Because the great number of patients in local Health Care System, really this paper looks very interesting.
The article is well written and explained. When I speak in my comment about “any ethic consideration” I mean that in the paper the authors don’t take in consideration the real satisfaction of the patients who are the protagonists of the study. When a group of scientists decide to front a political/social experiment, I suppose, they have to value also the impact they can reach on human emotionality and not only on the economy of the system. But in the last times, we are living, this is considered really not so important respect the reached good result in economy and efficiency. The question is: what we are really targeting in health policy for first? Good results in economy or good results in human satisfaction?
So this paper, I repeat good explained and written, make to take in ethic consideration this last aspect despite the well done job from sociological/scientific point of view.
I think it can be published as it is.
Authors response: Thank you for your comments. We feel that reviewer and us have reached the agreement on the interpretation of our findings. We think the good results for economy is only the intermediate output health policy and good results in human satisfaction and health is the ultimate goal of our health system.

Reviewer 3 Report
An important paper that shows that health system reform can happen in the capital city relatively rapidly with strong leadership and support from the central government.
In terms of limitations, could this restructuring have been so rapidly successful outside of the municipality of Bejing? Many LMIC would not have the number of health care providers available in Bejing. What might limitations be in much more rural areas of China? How easy would it be to improve drug availability in a far western region of China, for example?
Consider if any tables or figures could be "Supplementary".
There are a number of grammar mistakes: subject and verb agreement, appropriate verb tense, possessives, extra words, etc.
L. 125 - "To analysis" instead of "To analyze"...
L. 137 ..analysis was used "to the" pre- and post-reform data.
- 156. Why was a p<0.10 defined as significant?
Line 160. Consider use of COREQ checklist for reporting qualitative research…..See Tong A et al. 2007. International Journal for Quality in Health Care 6:349-357.
Table1. “mark-up”
Line 126-131. Clarify. Use at least two sentences.
Line 151-152. Incomplete sentence.
Line 172-173. Incomplete sentence.
- 182 & 183. “Last year”…….or “the previous year”?
Line 274. What is the meaning of “docking” in this context?
Line 365. In previous ??
Line 378. Healthcare-seeking behaviors.
Line 378. What is the meaning of “knobs” in this context.
Line 379-381. Rewrite sentence for clarity.
Line 400. Because of the “lack”…
Author Response
Response to reviewers
Dear editor,
Thank you very much for organizing the comments on our manuscript. We have revised the manuscript according to the editorial requirements and the comments from reviewers. Please find the responses to the Comments and Editorial Requirements below.
Comments:
An important paper that shows that health system reform can happen in the capital city relatively rapidly with strong leadership and support from the central government.
-In terms of limitations, could this restructuring have been so rapidly successful outside of the municipality of Bejing? Many LMIC would not have the number of health care providers available in Bejing. What might limitations be in much more rural areas of China? How easy would it be to improve drug availability in a far western region of China, for example?
Authors response: Thank you for your helpful suggestion. We agreed with your opinion. Improving the drug availability may be difficult for less developed regions. We have added it as limitation fourth in the article.
Consider if any tables or figures could be "Supplementary".
Authors response: Maybe table 2 could be “Supplementary”
There are a number of grammar mistakes: subject and verb agreement, appropriate verb tense, possessives, extra words, etc.
- 125 - "To analysis" instead of "To analyze"...
- 137 ..analysis was used "to the" pre- and post-reform data.
Authors response: Thank you for your kind reminder. We have corrected it in the manuscript. Line 139 &150
- Why was a p<0.10 defined as significant?
Authors Response: In our study, we summed the number of service volumes of each level institutions and divided by the number of institutions. Thus there are twelve points before and after the reform, respectively. In other words, the sample is relative small in our study. We defined p<0.10 as significant, which is widely used in the impact evaluation analysis, especially in the economic fields.
Line 160. Consider use of COREQ checklist for reporting qualitative research…..See Tong A et al. 2007. International Journal for Quality in Health Care 6:349-357.
Table1. “mark-up”
Authors Response: Thank you for your helpful suggestion. We have correct the description of the qualitative analysis according to the COREQ and added it as a reference in the manuscript. Line 172-186
Line 126-131. Clarify. Use at least two sentences.
Authors response: Thank you for your kind reminder. We have corrected it in the manuscript. Line:141-144
Line 151-152. Incomplete sentence.
Authors response: Thank you for your kind reminder. We have corrected it in the manuscript. Line165-166
Line 172-173. Incomplete sentence.
Authors response: Thank you for your kind reminder. We have corrected it in the manuscript. Line:200
182 & 183. “Last year”…….or “the previous year”?
Authors response: Thank you for your kind reminder. We have corrected it in the manuscript. Line:209&211
Line 274. What is the meaning of “docking” in this context?
Authors response: The docking in this context mainly referred to the drug catalogues were adjusted and the drugs for the four non-communicable diseases were the same between PHC and tertiary hospitals, which increasing availability of more essential medicines for non-communicable diseases.
Line 365. In previous ??
Authors response: Thank you for your kind reminder. We have corrected it in the manuscript. Please see Line:402
Line 378. Healthcare-seeking behaviors.
Authors response: Thank you for your kind reminder. We have corrected it in the manuscript. Line:416
Line 378. What is the meaning of “knobs” in this context.
Authors response: Knobs in this context referred to the policy handles which could be used to change the patients healthcare-seeking behaviors.
Line 379-381. Rewrite sentence for clarity.
Authors response: Thank you for your kind reminder. We have corrected it in the manuscript as follows: Line 417-419
Based on this, in the future policy design of public hospital reform, what we should remember is that the healthcare system is an interdependency macrocosm, and systematic thinking is of great importance to maximize the policy effect through an organic policy combination.
Line 400. Because of the “lack”…
Authors response: Thank you for your kind reminder. We have corrected it in the manuscript.
